# Interlaboratory Performance of a Real-Time PCR Method for Detection of *Ceratocystis platani*, the Agent of Canker Stain of *Platanus* spp.

**DOI:** 10.3390/jof8080778

**Published:** 2022-07-26

**Authors:** Angela Brunetti, Kurt Heungens, Jacqueline Hubert, Renaud Ioos, Gian Luca Bianchi, Francesca De Amicis, Anne Chandelier, Sietse Van Der Linde, Ana Perez-Sierra, Valeria Gualandri, Maria Rosaria Silletti, Vito Nicola Trisciuzzi, Silvia Rimondi, Tiziana Baschieri, Elio Romano, Valentina Lumia, Marta Luigi, Francesco Faggioli, Massimo Pilotti

**Affiliations:** 1Council for Agricultural Research and Economics, Research Centre for Plant Protection and Certification (CREA-DC), 00156 Rome, Italy; angela.brunetti@crea.gov.it (A.B.); valentina.lumia@crea.gov.it (V.L.); marta.luigi@crea.gov.it (M.L.); francesco.faggioli@crea.gov.it (F.F.); 2Flanders Research Institute for Agriculture, Fisheries and Food (ILVO), 9940 Merelbeke, Belgium; kurt.heungens@ilvo.vlaanderen.be; 3Plant Health Laboratory for the French Agency for Food, Environmental and Occupational Health & Safety (ANSES) Mycology Unit, 54220 Malzéville, France; jacqueline.hubert@anses.fr (J.H.); renaud.ioos@anses.fr (R.I.); 4Agenzia Regionale Per lo Sviluppo Rurale—ERSA Servizio Fitosanitario e Chimico, Ricerca, Sperimentazione ed Assistenza Tecnica, Struttura Stabile Laboratorio di Fitopatologia e Biotecnologie, Pozzuolo del Friuli, 33050 Udine, Italy; gianluca.bianchi@ersa.fvg.it (G.L.B.); francesca.deamicis@ersa.fvg.it (F.D.A.); 5Walloon Agricultural Research Centre—CRA-W Life Sciences Department, 5030 Gembloux, Belgium; a.chandelier@cra.wallonie.be; 6Forest Research Tree Health Diagnostic & Advisory Service, Alice Holt Lodge, Farnham, Surrey GU10 4LH, UK; s.vanderlinde@nvwa.nl (S.V.D.L.); ana.perez-sierra@forestresearch.gov.uk (A.P.-S.); 7FEM-IASMA, Centro Trasferimento Tecnologico Dipartimento Sperimentazione e Servizi Tecnologici, Unità Protezione Piante e Biodiversità Agroforestale, S. Michele all’Adige, 38098 Trento, Italy; valeria.gualandri@fmach.it; 8Centro di Ricerca, Sperimentazione e Formazione in Agricoltura, Basile Caramia, Locorotondo, 70010 Bari, Italy; mariarsilletti@crsfa.it (M.R.S.); nik.tris@gmail.com (V.N.T.); 9Servizio Fitosanitario Regione Emilia-Romagna, 40129 Bologna, Italy; silvia.rimondi@regione.emilia-romagna.it (S.R.); tiziana.baschieri@regione.emilia-romagna.it (T.B.); 10Council for Agricultural Research and Economics, Research Centre for Engineering and Agro-Food Processing (CREA-IT) Treviglio, 24047 Bergamo, Italy; elio.romano@crea.gov.it

**Keywords:** *Ceratocystis platani*, *Platanus*, canker stain, test performance study, Real-Time PCR, EvaGreen, SYBR Green, Taqman, performance parameters

## Abstract

*Ceratocystis platani* (CP), an ascomycetous fungus, is the agent of canker stain, a lethal vascular disease of *Platanus* species. *Ceratocystis platani* has been listed as a quarantine pest (EPPO A2 list) due to extensive damage caused in Southern Europe and the Mediterranean region. As traditional diagnostic assays are ineffective, a Real-Time PCR detection method based on EvaGreen, SYBR Green, and Taqman assays was previously developed, validated in-house, and included in the official EPPO standard PM7/14 (2). Here, we describe the results of a test performance study performed by nine European laboratories for the purpose of an interlaboratory validation. Verification of the DNA extracted from biological samples guaranteed the high quality of preparations, and the stability and the homogeneity of the aliquots intended for the laboratories. All of the laboratories reproduced nearly identical standard curves with efficiencies close to 100%. Testing of blind-coded DNA extracted from wood samples revealed that all performance parameters—diagnostic sensitivity, diagnostic specificity, accuracy and reproducibility—were best fit in most cases both at the laboratory and at the assay level. The previously established limit of detection, 3 fg per PCR reaction, was also validated with similar excellent results. The high interlaboratory performance of this Real-Time PCR method confirms its value as a primary tool to safeguard *C. platani*-free countries by way of an accurate monitoring, and to investigate the resistance level of potentially canker stain-resistant *Platanus* genotypes.

## 1. Introduction

Canker stain is a destructive disease of the *Platanus* species, caused by the ascomycetous fungus *Ceratocystis platani* (J. M. Walter) Engelbr. and T. C. Harr., a host-specialized species within the Latin American Clade (LAC) of the genus *Ceratocystis* sensu stricto [1,2,3]. The pathogen colonizes the roots and the stem of the trees by entering through the wounds or exploiting root anastomosis between diseased and healthy trees, and it causes wood discoloration and vessel dysfunction as well as necrosis of the cambium and living bark tissues thus resulting in the death of the trees [4,5]. Spread of *Ceratocystis platani* to disease-free areas can occur in a variety of ways. Foot and vehicle traffic, birds, rodents, and insects can transport either *C. platani*-infected sawdust from pruning of infected trees or fungal propagules that first form in the cracks between necrotized bark and wood and subsequently wash down to the base of the tree. Fungal inoculum from infected trees growing on river banks can be transported by the water flow, infecting healthy trees on the banks downstream [6]. Wind can also contribute to spread fungal propagules making possible new infection events if fresh pruning wounds are encountered [7]. Finally, *Platypus cylindrus,* a xylofagous frass-producing beetle, has been shown to be a vector of *C. platani* [8].

Native to the southeastern USA, this fungus has spread to several countries in Southern Europe and the Mediterranean basin where it is destroying urban and natural stands of highly susceptible *Platanus* species, *P. × acerifolia* (Ait.) Willd. and *P. orientalis* L. [3,5,9,10].

Due to the damage caused, *C. platani* was included by the EU in the list of quarantine pests included in the Annex II part B (Pests known to occur in the Union territory) of the COMMISSION IMPLEMENTING REGULATION (EU) 2019/2072 [11] and in the EPPO A2 list of quarantine organisms [12,13,14]. The A2 list contains by definition pests that are not widely distributed in the EPPO region and in fact the pathogen seems to be absent in northern Europe, and its presence in the African Mediterranean countries is only suspected. However, the pathogen could easily spread to Northern European countries, especially in a warming European climate.

In Italy, where *C. platani* is widespread, a ministerial decree established mandatory monitoring, prophylaxis, and eradication measures to safeguard plane trees [15,16].

The core of the EPPO mission is to protect plant health in agriculture, forestry, and the uncultivated environment through the development of international strategies and standards aimed at impeding the introduction and the spread of dangerous pests. Development and standardization of reliable diagnostic methods is essential to protect plant health. In this context, robust validation data is a requirement before any diagnostic method can be included in the official diagnostic EPPO protocols. Since 2014, EPPO hosts Euphresco (EUropean PHytosanitary RESearch COordination), a network of organizations (e.g., National Plant Protection Organizations (NPPO) and research institutes) that fund and coordinate research in plant health. Thus, Euphresco strongly encourages test performance studies (TPS) for interlaboratory validation of diagnostic protocols.

Traditional detection of *C. platani* has relied on a number of assays—microscopic observations, isolation on nutritive media, moist chamber, carrot test, and baiting test—which are time-consuming and prone to false negative response [14,17,18]. Recently Real-Time PCR based methods for *C. platani* detection were developed [7,17], representing the first rapid and reliable tools for the protection of *Platanus* spp.

Luchi et al. [7] designed a Real-Time PCR method specifically for monitoring and quantifying aerial inoculum of the pathogen. This method can be used to monitor pathogen levels during sanitation cuttings and to investigate airborne transmission.

Pilotti et al. [17] developed a Real-Time PCR method for in-wood detection, which is appropriate for diagnostic purposes as well as investigation of host resistance. This method includes three ITS1-targeted assays based on EvaGreen and SYBR Green dyes and a Taqman probe. These assays, validated in depth by in-house comparisons, showed high-level performance on several parameters: amplification efficiency, analytical specificity (inclusivity and exclusivity), diagnostic sensitivity and specificity, analytical sensitivity, enforceability in necrotic wood, and inter-operator, intra-laboratory reproducibility [17,18].

In this study, we report in detail the results of a TPS performed by nine European laboratories which provides further validation of the method by Pilotti et al. [17]. The EPPO guidelines for validation of diagnostic methods were used to design the TPS and calculate the performance criteria [19,20,21]. Overall, the TPS confirmed (nearly) best fit performance parameters for this method across laboratories. This shows that it is a reliable tool (i) to safeguard *C. platani*-free countries and zones by way of an accurate monitoring and (ii) to characterize resistance levels of potentially canker stain-resistant *Platanus* genotypes.

## 2. Materials and Methods

### 2.1. Wood Sample Collection and Assessment of Their Infectious Status

Wood samples were collected in 2017 from 15 plane trees. Trees were selected according to four different ranks (Table 1): (i) healthy trees (*n* = 3)—H; (ii) trees naturally infected with *C. platani* (*n* = 3)—NI; (iii) trees artificially infected with *C. platani* that were dead from a variable number of months, (*n* = 6)—AI [ these potted trees were inoculated at age four in April 2016 with tooth pick technique [4] and were killed by the disease in the same year]; and (iv) trees naturally affected by canker diseases other than canker stain (*n* = 3)—D. With regard to the latter rank two trees were affected by the typical cankers associated with *Fomitiporia mediterranea* (M. Fischer) and showed the typical fruit body of this hymenomycete [22] (referred to as D.1 and D.2 in Table 1). The third tree (D3) was affected by pruning wound-associated black/carbonaceous perennial cankers. All sampled trees were located in Rome, except for two NI trees located in Pisa.

All trees labelled as NI and D were located in urban avenues and suburban roads. They were all still viable though declining at the time of sampling. In all diseased trees, discolored wood was collected from the reaction zone.

For comparison purposes, all wood samples were tested using the following diagnostic non-molecular assays to assess the presence/absence of *C. platani*: (i) microscope assay, (ii) isolation on nutritive medium, (iii) carrot assay, (iv) moist chamber, and (v) bait-plant assay. The specific nature of perithecia and mycelium obtained in the various assays was ascertained by microscopic observations. This enabled the identification of *C. platani* conidia and ascospores for morphological diagnosis [4,14]. Details on microscope and biological diagnostic assays can be found in Appendix A.

### 2.2. DNA Extraction

Fungal mycelium for DNA extraction was obtained by scraping the mycelium of a *C. platani* isolate (C.P. 32) with a sterile pipette tip from actively growing colonies cultured on PDA (Potato dextrose agar, Oxoid-Unipath Ltd., Basingstoke, Hampshire, England). The wood and mycelium extractions started from aliquots of 700 mg. Fungal mycelium and wood tissue aliquots were ground to powder with liquid nitrogen using a sterile pestle and mortar, and DNA was extracted using a DNeasy Plant Maxi Kit (Qiagen, Hilden, Germany), following the manufacturer’s instructions. After the final elution, 800 µL flow-through was obtained. For samples under 800 µL (anyway not less than 790 µL), additional flow-through obtained from the second elution step was added to make up the difference in volume. DNA was quantified using a Thermo Scientific NanoDrop™ 1000 Spectrophotometer.

DNA samples obtained from the wood and fungal mycelium as described above are called ‘DNA-stock-samples’ (Dss) (Table 1 and Table 2).

### 2.3. The Participating Laboratories (PL), the Real-Time PCR Methods, Master Mixes, and Cycling Protocols

The participating laboratories (PL) were European research institutions and Italian phytosanitary services, all with a proven expertise in the use of Real-Time PCR for diagnostic purposes. The OL also participated in the TPS (Appendix A).

The focus of this interlaboratory comparison was the assessment of reproducibility and performance parameters of the Real-Time PCR method based on EvaGreen, Taqman, and SYBR Green variants [14,17,18].

Each PL chose to perform between two options: (i) one assay type (Taqman or EvaGreen or SYBR Green) with a Bio-Rad master mix and another commercial master mix specifically suited for Real-Time PCR; and (ii) two assay types using Bio-Rad master mixes. Bio-Rad master mixes were SsoFast™ EvaGreen^®^ Supermix (for EvaGreen assay), SsoAdvanced™ Universal Probes Supermix (Taqman assay), and SsoAdvanced™ Universal SYBR^®^ Green Supermix (SYBR Green assay). Master mixes by other companies were qPCR MasterMix and qPCR MasterMix No ROX (Eurogentec, Seraing, Belgium), Maxima Probe/ROX qPCR Master Mix and Maxima SYBR Green/ROX qPCR Master Mix (Thermo Fisher Scientific, Waltham, MA, USA), and Power SYBR Green PCR Master Mix (Applied Biosystem, Thermo Fisher Scientific).

In all the assays the concentration of the primers and, when used, the probe was kept fixed at 0.5 µM and 0.3 µM for each primer and probe, respectively. Use of Bio-Rad master mixes and the CFX96^TM^ thermocycler implied the integral application of the cycling protocol [14,17,18]: iinitial denaturation at 96 °C for 3 min; 40 cycles at 95 °C for 10 s (denaturation) and 66 °C for 20 s (annealing/extension); final extension at 72 °C for 5 min (the latter step was used exclusively for EvaGreen and SYBR Green assays). Slight changes to the amplification program were made according to the operating instructions of the commercial master mix used and to the thermocycler type. Specifically, the PL that choose master mixes other than Bio-Rad performed a preliminary test aimed at comparing different annealing/extension times—20 s (the canonical value) vs. 30, 40, and 60 s—using PAC and NAC (positive and negative amplification control). PAC was tested both undiluted and at a 1:10.000 dilution.

See Appendix A for details of the Real-Time PCR procedures, i.e., primers and probe sequences, composition of master mixes, reaction assembly, and thermal cycling conditions.

A cycle threshold (C*t*) of ≤37 was suggested by the OL and was applied in the TPS as the upper limit for a positive detection. In fact, it was previously established that 3 fg of *C. platani* gDNA was the lowest detectable DNA dose per PCR reaction (Limit of Detection, LoD), in the approximate C*t* range of 33–36, depending on the assay [18].

The fluorophores used were EvaGreen, SYBR Green I, 6-FAM (6-carboxy-fluorescein). The Real-Time PCR systems used were CFX96^TM^ (Bio-Rad), StepOne Plus^TM^ (Applied Biosystem, Thermo Fisher Scientific), Rotor-Gene ^TM^ 6000 (Corbett, Life Science), Lightcycler 480 (Roche), and Applied Biosystems^®^ 7500 (Applied Biosystem, Thermo Fisher Scientific). In all Real-Time PCR systems, the threshold line was set in an auto-calculating modality. Details on fluorophores and Real-Time PCR systems are given in Appendix A.

### 2.4. Diagnostic Confirmation of DNA-Stock-Samples (Dss), Aliquoting, Homogeneity Testing, and the Shipping Material

Dss were tested by the OL in triplicate with EvaGreen and Taqman assays (Bio-Rad master mixes) in order to confirm diagnosis performed with non-molecular assays.

The root square of the Dss ($\sqrt{16}$= 4) was tested for repeatability with 11 replicates in the EvaGreen assay. The Dss tested for repeatability were NI.3 and AI.6 (*C. platani*-positive) and D.1 and H.3 (*C. platani*-negative).

The 16 Dss were then aliquoted by distributing a volume, for each micro tube and for each participant, that was large enough to perform the homogeneity test by the OL and all the activities planned for the PL (Table 2 and Appendix A). The quantity was adjusted to cope with failures of each activity due to common human errors. These were named DNA-aliquot-samples (Das) and they represented the material of the *sensu strictu* TPS, based on blind samples.

The final panel test consisted of 20 test samples (Table 1 and Table 2):-16 blind test samples—15 DNA extracts from wood, and one from an axenic *C. platani* culture.-2 controls: PAC and NAC obtained from NI.3 and H.1 (also supplied as blind Das).-1 positive wood extract sample for the standard curve experiments (obtained from NI.2, also supplied as a blind Das), labelled DNA-aliquot-St.Cu. (Da.St.Cu.)-The gDNA from an axenic culture of *C. platani* for use in the analytical sensitivity test (also supplied as a blind Das)

Some Das were tested in triplicate with the EvaGreen assay to ascertain the homogeneity of Das among themselves and to compare them to the Dss from which they had been derived. A number of Das higher than the root square of their total number was tested (Table 2). We established that homogeneity was basically achieved if Dss and their relative Das gave the same qualitative diagnostic response—positive or negative. It was considered to be fully achieved if they also gave fluorescent signals at the same or very close C*t*.

All material—DNA aliquots, Bio-Rad master mixes, oligonucleotides, and the probe—was dispatched in polystyrene boxes of 15.4 cc volume filled with pellets of dry ice. The Real-Time PCR master mixes from companies other than Bio-Rad were also used by the PL.

### 2.5. Stability Test

A stability test was performed by the PL on PAC and NAC in order to test the stability of both DNA samples and reagents after shipping. Each control was tested in triplicate with the Bio-Rad master mixes that had been shipped together with the PAC and the NAC. In addition, the OL tested all Dss 6 months after the completion of the TPS. The full compliance of the stability test was based on the same quali-quantitative criterion ruling the homogeneity test.

### 2.6. The Test Performance Study with Real-Time PCR: (i) Generating Standard Curves, (ii) Testing Blind-Coded DNA-Aliquot-Samples (iii) Testing Analytical Sensitivity

The TPS was executed by the PL and the OL.

In order to assess the reproducibility of the amplification efficiency of the Real-Time PCR on wood extracts, the PL and the OL generated standard curves with Da.St.Cu. In addition to the undiluted DNA extract (1:1), six five-fold serial dilutions were used (1:5, 1:25, 1:125, 1:625, 1:3.125, 1:15.625) and each was tested in triplicate using 2 μL per PCR reaction. PL performed one or two real-time PCR assays chosen among Taqman, EvaGreen, or SYBR Green versions, and using Bio-Rad and/or other commercial master mixes. The OL performed all three assays.

Blind Das were tested in triplicate.

Taqman assay was primarily used to test the analytical sensitivity but EvaGreen and SYBR Green assays were also used by some laboratories. The OL instructed PL to dilute the aliquot sample containing *C. platani* gDNA to obtain 7.5 and 1.5 fg per μL in the last two dilutions. Two microlitres for each dilution were then used to test the capacity of the method to detect 15 and 3 fg *C. platani* gDNA per PCR reaction. Each DNA quantity was tested in six replicates.

The complete list of the codes of the PL and the OL with the relative Real-Time PCR assay, master mix, and Real-Time PCR apparatus used is reported in Table 3, where each letter identifies a PL and the number the type of test performed. A chronological listing of scheduled TPS actions can be found in Appendix A.

### 2.7. Performance Criteria, Nomenclature, and Statistical Analysis

The Real-Time PCR assays were evaluated for their capacity to produce accurate and reproducible results for the detection of *C. platani*. We thus inferred the following performance parameters from TPS qualitative results: accuracy (AC), diagnostic sensitivity (DSE), diagnostic specificity (DSP), repeatability (‘accordance’) (DA) and reproducibility (‘concordance’) (CO) (Table 4).

AC is the closeness of agreement between detection results of the laboratories and the assigned values for the samples (i.e., their true sample status, infectious and healthy). AC is based on DSE and DSP. DSE is a measure of the ability of the method to detect the target in samples in which the target is present, i.e., those with a positive assigned value (N^+^), whereas DSP is the capacity of the method to fail the detection in samples in which the target is absent, i.e., those with a negative assigned value (N^−^). DA is the level of agreement between replicates of a sample tested under the same conditions; in this work, we calculated DA on results from testing Dss with 11 technical replicates (performed by OL) and the C.P. 32 gDNA with six technical replicates (analytical sensitivity). CO is the ability of the method to provide consistent results when applied to aliquots of the same sample tested under different conditions (i.e., by different laboratories). All definitions and calculations were taken from ISO 16,140 [23], OEPP/EPPO standards [19,20,21], Chabirand et al. [24,25], and Langton et al. [26]. In this work the positive and the negative detections obtained for samples for which the assigned value is, respectively, positive (infected) and negative (healthy) are referred to as true positives (TP) and true negatives (TN). False positives (FP) are results obtained for samples where a negative result is expected (given the true sample status); similarly, false negatives (FN) are results obtained for samples where a positive result is expected.

Calculations used are listed in Table 4.

The qualitative input data were the positive and the negative detections derived from each technical replicate from testing the DA of Dss, the stability of the shipped material and the analytical sensitivity. For blind Das, the qualitative result was considered both at (i) the technical replicate level; and (ii) the Das level, i.e., positivity and negativity was attributed to a Das by coherent results obtained from at least two technical replicates (out of three).

The dataset is a binomial distribution, thus the *binconf* function of the *Hmisc* package of R [27] statistical software was applied to search for confidence intervals (95%) for AC, DSE, and DSP criteria, using the Wilson score method with no continuity correction [28]. Tests on the equality of AC, DSE, and DSP between laboratories, and the chemistry-based assays were performed using a one-sample t-test.

Standard curves were compared according to the following procedure: the average value at each dilution point was subtracted from the corresponding values of a theoretical curve sharing the template values in the x axis (expressed in base-10 logarithm) with experimental curves, but with an exact best fit efficiency (100%) (i.e., with a regression equation with 3.32 as the angular coefficient). We thus obtained seven normalized replication values for each curve, which were used in the statistical analysis to compare the various curves and to give a measure of the significance among them. Fisher’s test for the analysis of variance (ANOVA) was developed on the dataset obtained after verifying the normality of the distribution with the Shapiro–Wilk test and the homogeneity of the variances with the Bartlett test.

### 2.8. Outliers

Data sets were considered outliers and were excluded from analysis when: (a) results of controls were non-concordant; (b) accuracy was statistically different from the average of accuracy obtained by all laboratories (n. FN or FP > average FN or FP ± 3 SD); (c) results of one test were incomplete (e.g., no technical repetition reported); (d) the number of undetermined results was significantly different from the other laboratories (n. undetermined/inconclusive > average undetermined ± 3 SD) [29].

### 2.9. Disclosure Policy by the Organiser

After contacting PL, OL distributed the ‘TPS plan’ describing the project. After acceptance by PL, OL also sent the ‘operational instructions’ on how to perform all the tests.

The 16 blind samples (Das) were labelled with numbers from 1 to 16 followed by the letter representing each PL. Numbering of Das was randomly attributed and was different among all PL. Only OL kept the correspondence between the Das of each PL and the Dss. To maintain the confidentiality of the link between each PL and its own results, we use the letter assigned to the PL (‘A-H’) in this paper as well. The results of the OL (‘Z’) are made public.

## 3. Results

### 3.1. Sample Preparation

The OL collected the wood samples for the TPS from 15 plane trees and verified their status (infected/healthy). Diagnosis performed with non-molecular assays confirmed the presence of *C. platani* in those samples collected from trees affected by canker stain-like symptoms and from trees artificially infected with *C. platani*. The pathogen was not detected in samples of healthy-looking non-inoculated trees, and from trees of urban avenues affected by disease other than canker stain (Table 1). For details on the outcome of non-molecular assays, see Appendix A.

Testing of the DNA-stock-samples (Dss), which had been extracted from the above reported trees with Real-Time PCR, confirmed the diagnosis performed with non-molecular assays. Melting peaks were detected in all *C. platani*-infected Dss, with a melting temperature of 81–81.5 °C as expected [18]. No peaks were detected in the *C. platani* uninfected Dss, confirming that the corresponding wood samples/trees were free of *C. platani*. The absence of peaks with a melting temperature other than that of *C. platani* target amplicon also confirmed the absence of primer dimers or other aspecific amplification products (Figure 1). The coherence between traditional and molecular diagnosis thus verified the real infectious/healthy status of the TPS samples.

Repeatability (DA) of detection was also fully confirmed with 11 replicates in four selected Dss (out of 16) that reached 1, i.e., a perfect fit. Figure 2a illustrates the comparison between first Real-Time PCR detections on Dss (three replicates) and detections for DA. Samples not infected with *C. platani* yielded no fluorescent signals.

Dss were thus used to make Das intended for the PL. Homogeneity was tested in selected Das (Table 2) and was fully confirmed comparing with the quali-quantitative response of the Dss and obtaining highly similar C*t* values (Figure 2b). The ranges of variation were all between 0.08 and 0.29 C*t*.

The stability test was performed by the PL to assess that material received by post from the OL remained unaltered after shipping. All tests gave the correct qualitative response, with full DA and similar C*t* values in the PAC, i.e., with a range of variation among the average values from the different PL of 1.52 C*t* in the EvaGreen assay and of 5.3 C*t* in the Taqman assay (Figure 2c,d). Testing all Dss in EvaGreen after the completion of the TPS again confirmed their stability as average detections for each Dss varied between 0.6–1.5 C*t* in comparison with those performed at the beginning of the preparatory phase.

### 3.2. Thermal Cycling Adjustments

The Real-Time PCR method was developed using Bio-Rad master mixes [17]. Use of different master mixes implied that slight changes to the amplification program were made according to the product instructions and a preliminary test aimed at identifying the optimal annealing/extension times. In these tests, 60 s was chosen for Eurogentec qPCR MasterMix and qPCR MasterMix No ROX to perform subsequent experiments, 30 s for Thermo Scientific Maxima Probe/ROX qPCR Master Mix and Maxima SYBR Green/ROX qPCR Master Mix, and with Applied Biosystems Power SYBR Green PCR Master Mix.

A full description of the Real-Time PCR method for *Ceratocystis platani* detection used in this TPS by the OL and the PL are contained in Appendix A.

### 3.3. Test Performance Study: Generating Standard Curves

To validate reproducibility of Real-Time PCR efficiency, the OL and six PL generated a total of 20 standard curves using a wood extract DNA and different assays, master mixes and Real-Time PCR systems (Table 3). In all standard curves the most dilute replicate sample was detected. In general, efficiency and R^2^ were close to the best fit except for curves obtained with Taqman and EvaGreen assays in the Lightcycler 480 system, whose efficiency was 79.7 and 74.3, respectively. As this was the only case of low performance among all standard curves obtained, we considered these data as outliers and did not include the two curves obtained with a Lightcycler 480 system (D) in further comparisons among the curves. Table 5 lists the performance parameters for each standard curve.

Specifically, we evidenced the interlaboratory reproducibility by comparing all standard curves obtained with CFX96^TM^ and Bio-Rad master mixes both with Taqman and EvaGreen assays (Figure 3a,c, Table 5). Regarding the Taqman assay (five curves by five laboratories) efficiency and intercept values varied within the ranges 94.1–99.9 and 33.7–34.6 (i.e., a variation of 0.9 C*t*), respectively. The same parameters of curves obtained with EvaGreen (four curves by four laboratories) varied within the range 98.3–100.7 and 33.5–34.0 (i.e., a variation of 0.5 C*t*), respectively.

In the Taqman assay, we also compared all curves performed with any commercial master mix on the three different Real-Time PCR systems used. These comparisons showed that all curves performed with the same Real-Time PCR system, although with different master mixes, were quite close/overlapping. On the other hand, the three different Real-Time PCR systems determined three discrete parallel bundles of curves. Curves obtained with Rotor-Gene ^TM^ 6000 were the lowest and those obtained with CFX96^TM^ and StepOne Plus^TM^ were the highest and intermediate, respectively (Figure 3b). Statistical analysis revealed a significant difference among the Taqman curves obtained with the different Real-Time PCR systems (*p* < 0.05) (Figure 3b).

Four experiments were performed in SYBR Green assay, which used two different master mixes on CFX96^TM^ and two on Applied Biosystems^®^ 7500 (Table 3). Even these curves overlapped based on the thermocycler system, largely irrespective of the master mix. Curves obtained with CFX96^TM^ were the lowest (Figure 3d). In this case, the curves did not show statistically significant differences.

The box plots in Figure 4 show the variability of the data series; the distance between the boxes highlights the differences.

### 3.4. Test Performance Study: Blind Testing of DNA-Aliquot-Samples

All the participants laboratories were able to submit the results for a total of 20 data sets and 320 results from blind test items. Specifically, in all three assays—EvaGreen, SYBR Green, and Taqman—all Das which were N^+^ were always correctly detected, at both the technical replicate and the Das level, namely only TP and no FN were obtained. Thus, DSE was 100% at the laboratory and the assay level. A quantitative view of fluorescent detection signals (C*t*) obtained with positive Das is depicted in Appendix B, Figure A1. The Taqman assay performed by PL C with the Rotor-Gene ^TM^ 6000 thermocycler clearly revealed that this system ensured the lowest C*t* while keeping the highest DSP.

Similarly, all detections performed by the laboratories on N^−^ Das showed best fit except in the following cases: (i) six FP technical replications referring to two Das of the same experiment performed with SYBR Green assay; and (ii) two FP technical replications referring to two Das of the same experiment and performed with a Taqman assay (Table 6). In the latter case, the FP fluorescent signals were just below the threshold of C*t* 37, which suggests a very slight contamination. Note that such FPs do not condition the Taqman performance parameters at the level of Das (being one FP out of three technical replicates for each Das).

In summary we obtained 2 FP out of 198 technical replicate-detections of Das which were N^−^ in Taqman, no FP out of 90 N^−^ in EvaGreen, and 6 FP out of 72 N^−^ in SYBR Green. No FN were obtained out of 330, 150, and 120 technical replicate-detections of Das, which were N^+^ in Taqman, EvaGreen, and SYBR Green, respectively. Accordingly, values of AC and CO globally referred to each assay were either high or best fit.

Table 6 shows the value and the confidence intervals of the performance parameters. Presented values of performance parameters are those calculated based on the results at the level of technical replicates, i.e., the more stringent criterion. Appendix A reports the qualitative results detailed at the level of both technical replicate and Das, as well as the calculations made to determine the performance parameters.

### 3.5. Test Performance Study: Testing Analytical Sensitivity

Testing analytical sensitivity in EvaGreen and SYBR Green assays always yielded the best fit of performance parameters with both 15 and 3 fg (Table 7). In the Taqman assay, performance was at the highest level in 6 PL out of 8 with 15 fg, and in 4 PL out of 8 with 3 fg (Table 7). Specifically, (i) two FN replicates (referring to two tests) out of 48 N^+^ replicates were detected with 15 fg; and (ii) six FN replicates (referring to four tests) out of 48 N^+^ replicates were detected with 3 fg. However, all FN gave a fluorescent signal with C*t* values in the range 37.07–37.74 (slightly above the established threshold for a positive detection).

In conclusion, reproducibility (CO) of analytical sensitivity was 100% in EvaGreen and SYBR Green assays and 92.9 and 76.2% in Taqman assays with 15 and 3 fg, respectively (Table 7, Appendix A).

The confidence intervals are reported in Table 7 (the different values are verified for statistical significance for *p*-value < 0.001). Figure 5 shows a quantitative view of the fluorescent detection signals obtained (C*t*).

In this analysis, the experiment performed by D was not included as the default settings of the Real-Time PCR system they used—Lightcycler 480 (Roche)—does not permit *Ct* values over 35. This caused problems because the detection of 3 fg frequently was around this value in several PL and in the OL.

## 4. Discussion

This TPS, performed exclusively with laboratories proficient in Real-Time PCR, confirms the excellent performance of our intra-laboratory validation of the method [17,18]. The TPS was articulated in five phases: (i) preparation, (ii) the validation of the amplification efficiency through the inference of standard curves, (iii) blind detection of samples, (iv) validation of the analytical sensitivity, and (v) assessment of all validation parameters.

All verifications performed in the preparatory phase guaranteed that all material was compliant, i.e., (i) the infectious state of all samples had been correctly determined by integrating the results of different methods, (ii) the Dss were amplifiable and amplification results were fully repeatable, (iii) Das intended for the PL were homogeneous, and (iv) all dispatched material, samples, and reagents remained unaltered after delivery. Finally, the annealing/extension time of the cycling protocol was also successfully adjusted for master mixes other than Bio-Rad.

All three tested assays—EvaGreen, Taqman, and SYBR Green—resulted in (nearly) best fit data. Overall, the method has proven not only reproducible but also robust—hence transferable—that is to say that the performance did not substantially change when using different master mixes and Real-Time PCR platforms, which is not a foregone conclusion when testing a method in collaborative studies [30].

Analysis of the standard curves reveals that PL reproduced a high amplification efficiency and data linearity, obtaining values that approach a best fit. This also clearly suggests that the whole procedure—DNA extraction and Real-Time PCR—would not be affected by inhibitors, which are potentially present in necrotic wood infected with *C. platani*.

In the absence of a suitable method for the statistical comparison of the different standard curves, we developed a method that we present here for the first time. This method is based on normalizing the C*t* data of each dilution point of the curve with those of a theoretical best fit curve. This results in seven replicate values (corresponding to the seven dilution points) available for each curve to infer a mean and compare it with ANOVA. Interestingly, no significant differences could be found among the curves obtained when using the same Real-Time PCR system, regardless of the master mix used. This clearly suggests that use of the different master mixes with the adjusted thermal condition did not influence PCR efficiency nor the intercept value. This provides good evidence of both the reproducibility and the versatility/robustness of the method.

In contrast, significant differences were found among the curves obtained with the Taqman assay depending on the different Real-Time PCR systems that had been used. All of the curves were essentially parallel, namely with similar angular coefficient and thus efficiency; therefore, the intercept was responsible for the diversity in curve bundles. Specifically, the curves obtained with Rotor-Gene^TM^ 6000 had the lowest intercept, StepOne Plus^TM^ was intermediate, and CFX96^TM^ was the highest. This clearly shows how the Real-Time PCR systems that enabled inference of curves with lower intercept values guaranteed the detection of the same target DNA quantity—thus also of the LoD—at a lower C*t* than the other systems. The capacity of the Rotor-Gene^TM^ 6000 for detection at a lower C*t* than other platforms was also previously observed in a comparison with Lightcycler 480 [31,32]. This would also imply that Rotor-Gene^TM^ 6000 has the potential to lower the LoD of our method, which had been established using the CFX96 ^TM^ [18]. Indeed, in the analytical sensitivity test conducted by one PL using the Rotor-Gene^TM^ 6000, the LoD of 3 fg was detected at C*t* 31.6 on average (SD = 0.5), whereas in those conducted by six PL using the CFX96 detection was at C*t* 35.6 on average (SD = 1.3). This indicates that the Rotor-Gene^TM^ 6000 could theoretically enable the detection of a LoD ten times lower than 3 fg (i.e., 0.3 fg per PCR reaction) at a C*t* value roughly 3.3 cycles later than 31.6 (i.e., at C*t* 34.9), which is still allowed for a positive detection. Obviously, this has to be practically demonstrated as well as a demonstration that the thermocycler does not create false positive signals at a higher C*t*.

Regarding detection of blind samples, all three assays obtained either very high or best fit CO, as derived from the (nearly) best fit performance parameters obtained by each PL. In practice, we analyzed data starting either from results on technical replicates or from those at the Das level (i.e., those derived from in-agreement results from at least two replicates out of three). While the former is more stringent, the latter can mask any inconsistent results within the technical replicates, i.e., if one replicate out of the three is an FP or an FN. In Table 6, we present results obtained with the first method while in Spreadsheet S1 both are presented.

With regard to N^−^ samples (*n* = 360 at the technical replicate level), a total of eight FP (2.2% of all N^−^) were detected. Six of these were detected in SYBR green by a single PL, which had a substantial impact on the performance of the method by this PL. Nevertheless, diagnostic specificity (DSP), accuracy (AC), and CO of the assay remained high at 91.7, 96.9, and 93.2, respectively. Taqman and EvaGreen showed a best fit.

All N^+^ samples (*n* = 600 at the technical replicate level) were detected as positive (FN were 0% of all N^+^) and C*t* values among the different PL were very close, regardless of the master mix used. This again confirms the robustness of the method as stated above. To corroborate the significance among standard curves obtained in Taqman with the different Real-Time PCR systems, all detections performed with Rotor-Gene^TM^ 6000 (Taqman) were at a clearly lower C*t* than the others.

The analytical sensitivity of this Real-Time PCR method (LoD of 3 fg per PCR reaction in Taqman, EvaGreen, and SYBR Green assays) has been thoroughly validated by Lumia et al. [18] in an intra-laboratory performance test. The inter-laboratory comparison of this work again confirmed the capacity of the assays to detect 3 fg gDNA per PCR reaction. While all performance parameters and CO had best fit in the EvaGreen and SYBR Green assays, they were somewhat lower in the Taqman assay, where CO was 92.9 and 76.2 when testing 15 and 3 fg, respectively (Table 7). A possible explanation of this partial lack of homogeneity among the PL might be due to the fact that each PL performed in-house the dilutions of the gDNA sample dispatched by the OL. When performing a dilution, a minimal and an inadvertent variation in the first dilution steps can significantly impact the real value of the lowest concentration. This can thus condition the DA of the detection at what is believed to be the LoD, which by definition borders on undetectable DNA concentrations.

In diagnostic methods, validation and standardization of analytical sensitivity is key. Especially in highly sensitive methods such as this, validation of LoD is necessary in order to exploit the detection potential in studies where quantification is required. One example is the accurate evaluation of the repeatability/reproducibility of the resistance response of plants to pathogens with a specific focus on a very small pathogen titer, as this is frequently linked to the condition of resistance. We have previously shown that this Real-Time PCR method enabled a dramatic distinction between resistant and susceptible *Platanus* genotypes in terms of the target copy number of *C. platani* in the inoculated trees [17].

However, it should be considered that a genuine detection at the LoD can be indistinguishable from a contamination, which typically gives late fluorescent signals similar to detection of the LoD. It is therefore important that the Real-Time PCR plate is scattered with many replicates of a no-template or healthy control, as a guarantee of the accuracy of the results when they do not yield fluorescent signals. For routine diagnosis, use of controls is a must, but the number of replicates can be limited. Practical experience indicates that testing samples extracted from heavily infected wood can cause a contamination that spreads to the wells of the plate and can show up as late fluorescent signals. Thus, we recommend that in assays performed with diagnostic purposes all samples yielding late fluorescent signals from 34/35 C*t* upward should be either a) repeated with a consistent array of negative controls (no template controls and extracts from healthy wood) or b) repeated with new samples collected from tissue possibly affected by fresh pathological necrosis.

Real-Time PCR technology is still the gold standard for an early, reliable, accurate, robust, and sensitive pathogen detection. In fact, among its many applications, it is used extensively for high-throughput screening and early diagnosis of SARS-CoV-2 and other infectious human diseases [33]. However, newer but already established technologies such as third-generation sequencing technologies are now being considered for their diagnostic potential [34,35]. Developing new methods for *C. platani* detection by exploiting these technologies would represent a new challenge for diagnosis and research on canker stain of the plane tree. This approach could integrate the still unequalled diagnosis by Real-Time PCR with additional epidemiological data that could be obtained upon detection of intraspecific sequence variants.

## 5. Conclusions

The excellent results of the TPS confirmed this Real-Time PCR method as a primary diagnostic tool for monitoring *C. platani*-free zones. Thus, this method would be crucial to impede the spread of canker stain in Northern European areas, such as Austria, Germany, Belgium, and the United Kingdom, where the disease is not yet present but plane tree is widely grown in the urban environments. Moreover, this Real-Time PCR can confidently be used by all researchers working on the characterization of the resistance/susceptibility levels of *Platanus* genotypes to canker stain. Indeed, a consistent detection protocol will enable reliable comparisons among the results obtained in different selection programs.

## Figures and Tables

**Figure 1 jof-08-00778-f001:**
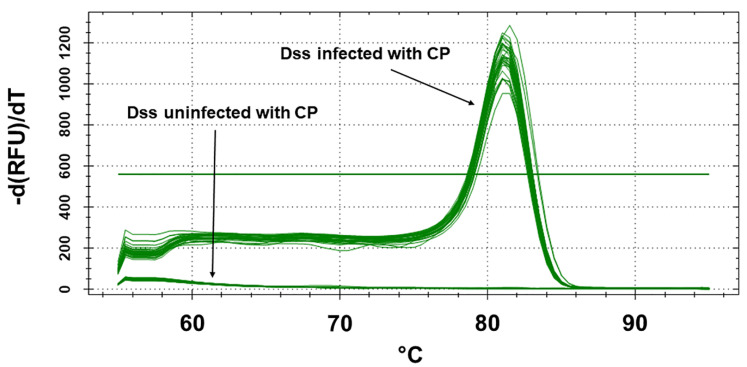
Melt-curve peaks inferred from the dissociation run following detection by the Real-Time PCR EvaGreen assay of DNA stock samples (Dss) extracted from wood samples and from an axenic culture of *C. platani* (C.P. 32). Dss were used to make the DNA-aliquot-samples (Das), which were submitted to the participating laboratories (PL) for blind detection.

**Figure 2 jof-08-00778-f002:**
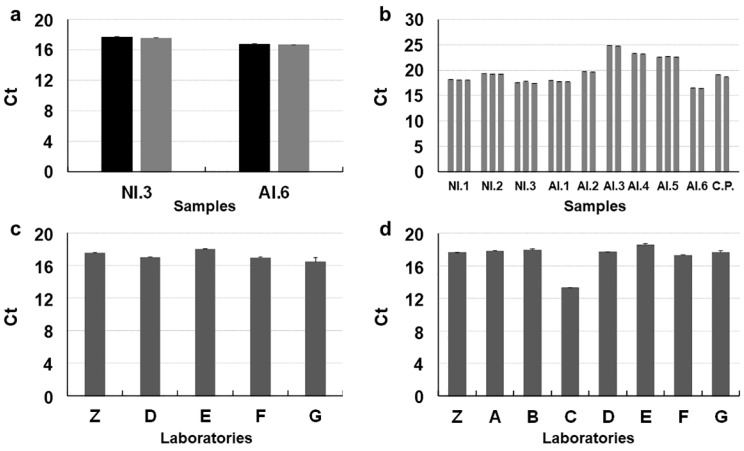
Preparatory phases of the test performance study (TPS) for diagnosis of *Ceratocystis platani* with Real-Time PCR. (**a**) Test of repeatability (DA) performed by the organizing laboratory (OL = Z) with EvaGreen assay on the root square of DNA stock samples (Dss) (*n* = 4); DA of the two *C. platani*-infected Dss is shown (the *C. platani*-uninfected Dss—D.1 and H.3—consistently gave no signal, data not shown); black columns represent the C*t* mean of 11 technical replicates and are compared with the *Ct* mean of three technical replicates (grey columns), which were obtained from the same Dss in the initial detection phase. (**b**) Homogeneity test (performed by OL): here we compare results of the EvaGreen assay from the initial detection phase on Dss (in triplicate, first column of each group) and from detection on the DNA-aliquot-samples (Das) (in triplicate) derived from Dss and intended for PL; nine Das from *C. platani*-uninfected samples were also tested and gave no signal, as expected (data not shown) (**c**,**d**) Stability test: results of testing a positive amplification control (PAC = NI.3) by the OL and the participating laboratories (PL) with Bio-Rad EvaGreen and Taqman assays, after they received from the OL the material, which included also the Bio-Rad Real-Time PCR master mixes. This test also included a negative amplification control (NAC = H.1), which was always negative (data not shown). The bars indicate the standard deviation of the mean.

**Figure 3 jof-08-00778-f003:**
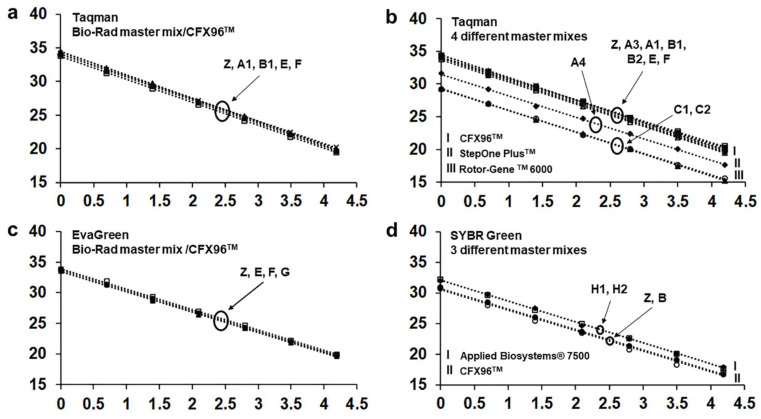
Standard curves performed by the organizing laboratory (OL = Z) and the participating laboratories (PL) (depicted with uppercase letters), with the different assays, commercial master mixes, and Real-Time PCR systems. *Ct* values are on the *y*-axis and the log base 10 of the dilution factor is represented on the *x*-axis. (**a**,**c**) curves performed by different PL with the same assay, master mix, and Real-Time PCR system show full interlaboratory reproducibility as quite close/overlapping (no significant differences, *p* > 0.05); (**b**) curves obtained in a Taqman assay with different commercial master mixes and different Real-Time PCR systems show that the latter factor always determines significant differences (*p* < 0.05); (**d**) curves obtained in SYBR green assay. A distinction according to the Real-Time PCR system is also evident but not significant. The number following the letter representing the PL indicates the master mix/Real-Time PCR system adopted (see Table 3).

**Figure 4 jof-08-00778-f004:**
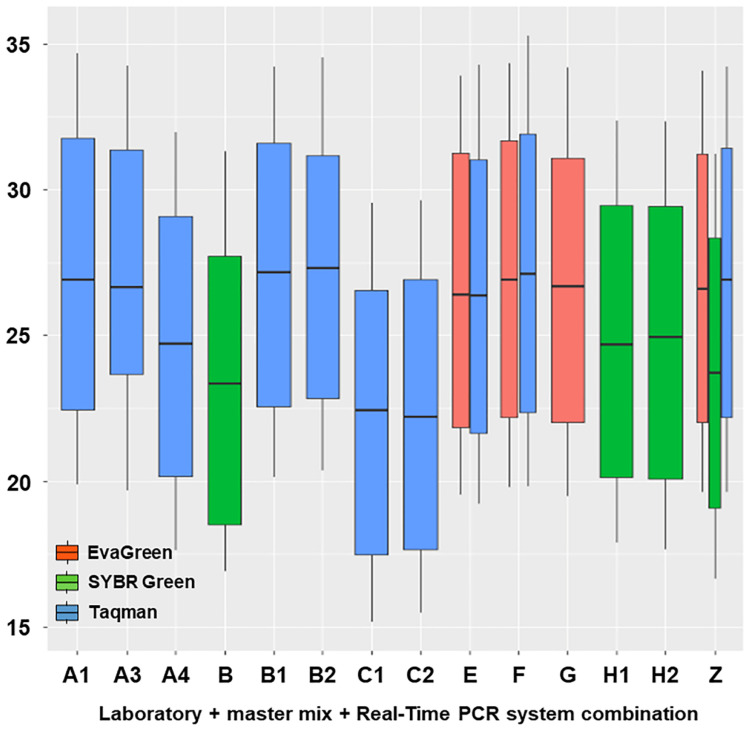
Box plot of the C*t* values of the standard curves carried out in the various laboratories and normalized to a theoretical best fitting curve. Curves were obtained with Taqman, EvaGreen, and SYBR Green assays. The organizing laboratory (OL = Z), participating laboratories (PL = A, B, C, E, F, G, H). For the combination laboratory + master mix + Real-Time PCR system, see Table 3.

**Figure 5 jof-08-00778-f005:**
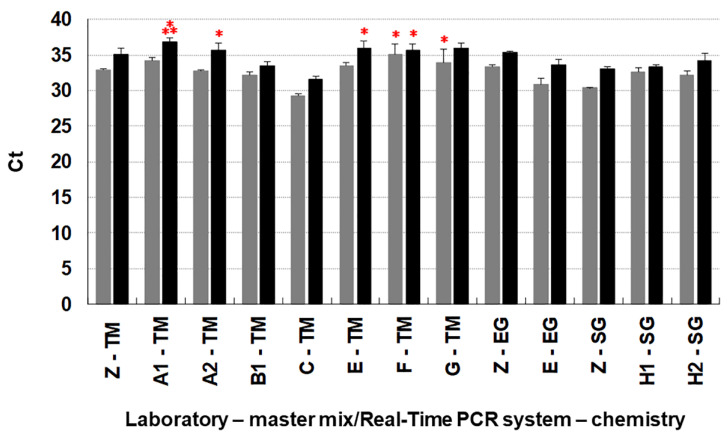
Analytical sensitivity tested in Taqman (TM), EvaGreen (EG), and SYBR Green (SG) assays by the organizing laboratory (OL = Z) and the participating laboratories (PL), represented by upper case single letters or alphanumeric codes to represent the adopted combination master mix/Real-Time PCR system (see Table 3). The black columns represent the mean *Ct* values at which the lowest detectable pathogen DNA quantity i.e., 3 fg gDNA of *Ceratocystis platani*, was detected, as previously reported by Lumia et al. [18]. Grey columns are the average detection *Ct* of a quantity of DNA five times larger, 15 fg. The bars are the standard deviation of six replicate-based means. The number of red asterisks above each column represent the number of false negatives (FN) obtained in each experiment.

**Table 1 jof-08-00778-t001:** List of the samples prepared for TPS, including blind samples and controls and their related expected diagnostic outcome. The phytosanitary state was ascertained using morphological/biological assays and Real-Time PCR.

Sample ID	Nature of the Sample	Number of Samples	Sample Type	Expected Detection
NI.1→ NI.3	*C. platani* Naturally-Infected tree	3	Blind samples	Positive
AI.1→AI.6	*C. platani* Artificially-Infected tree	6	Blind samples	Positive
H.1→H.3	Healthy tree	3	Blind samples	Negative
D.1→D.3	Diseased tree infected with non-target species	3	Blind samples	Negative
gDNA C.P. 32	Pure *C. platani* colony	1	Blind sample	Positive
NAC	Healthy tree (=H.1)	1	Negative amplification control	Negative
PAC	*C. platani* Naturally-Infected tree (=NI.3)	1	Positive amplification control	Positive
DNA-aliquot-St.Cu	*C. platani* Naturally-Infected tree (=NI.2)	1	Standard curve validation	Positive

**Table 2 jof-08-00778-t002:** DNA-stock-samples (Dss) used to perform the test performance study (TPS) and random blind coding of the corresponding aliquots intended for the organizing laboratory (OL) and the participating laboratories (PL—A to H). These aliquots are called ‘DNA-aliquot-samples’ (Das). All Das were blinded. Grey-shaded Das were checked by the OL using the Real-Time PCR EvaGreen assay in the homogeneity test. PAC and NAC (positive and negative amplification controls) were used by the PL for the stability test soon after receiving the TPS material, as well as for assessment of the best annealing/extension time for use of the master mixes other than Bio-Rad. Da.St.Cu. (DNA aliquot Standard Curve) was given to the PL to generate standard curves. *Ceratocystis platani* C.P. 32 gDNA was supplied in duplicate to the PL to perform the blind test and the analytical sensitivity test. + = aliquot provided, − = aliquot not provided.

	1	2	3	4	5	6	7	8	9	10	11	12	13	14	15	16	17	18	19	20
Dss ^1^	NI.1	NI.2	NI3	AI.1	AI2	AI3	AI4	AI5	AI6	H.1	H.2	H.3	D.1	D.2	D.3	gDNA C.P. 32	PAC (=NI.3)	NAC (=H.1)	Da.St.Cu. (=NI.2)	gDNA C.P. 32
Z (OL)	6	3	11	12	14	15	10	2	1	4	7	5	8	16	9	13	+	+	+	+
A	3	14	13	10	7	9	11	15	12	1	8	16	4	6	5	2	+	+	+	+
B	12	7	14	16	15	11	8	6	10	2	13	3	1	9	4	5	+	+	+	+
C	11	12	15	14	6	4	3	10	7	9	1	2	16	5	13	8	+	+	+	+
D	9	5	7	8	13	10	16	11	15	12	4	1	6	2	3	14	+	+	+	+
E	7	9	2	5	12	1	4	13	3	11	6	8	15	14	16	10	+	+	+	+
F	4	16	10	11	9	5	13	1	6	7	3	15	2	8	14	12	+	+	−	+
G	2	13	8	7	10	14	12	5	16	15	9	11	3	4	6	1	+	+	+	+
H	10	15	6	9	2	13	5	12	14	3	16	4	7	1	8	11	+	+	−	+

^1^ NI = *Ceratocystis platani*-Naturally Infected, AI = *Ceratocystis platani*-Artificially Infected, H = Healthy, D = Diseased tree non target.

**Table 3 jof-08-00778-t003:** Master mix/Real-Time PCR system combinations adopted in the test performance study (TPS) by the organizing laboratory (OL = Z) and the participating laboratories (PL, the other different letters). All combinations were used to obtain standard curves (except A2 and G in Taqman) and to detect the blind samples (except A3 and A4). Those tested for analytical sensitivity are marked with red asterisks. Numbering for each letter, when present, indicates that different master mix/Real-Time PCR system combinations were used by a PL with the same assay/chemistry. The names of Bio-Rad master mixes have been shortened, see the text for the full names.

OL, PL	Taqman	Eva Green	SYBR Green
Z	Universal Probes Supermix (BioRad) CFX96™ (BioRad) *	EvaGreen^®^ Supermix (BioRad) CFX96™ (BioRad) *	SYBR^®^ Green Supermix (BioRad) CFX96™ (BioRad) *
A1	Universal Probes Supermix (BioRad) CFX96™ (BioRad) *		
A2	qPCR MasterMix (Eurogentec) StepOne Plus^TM^ (Applied Biosystem) *		
A3	qPCR MasterMix (Eurogentec) CFX96™ (BioRad)		
A4	Universal Probes Supermix (BioRad) StepOne Plus^TM^ (Applied Biosystem)		
B			Maxima SYBR Green/ROX qPCR Master Mix (Thermo Scientific) CFX96™ (BioRad)
B1	Universal Probes Supermix (BioRad) CFX96™ (BioRad) *		
B2	Maxima Probe/ROX qPCR Master Mix (Thermo Scientific) CFX96™ (BioRad)		
C1	Universal Probes Supermix (BioRad) Rotor-Gene ^TM^ 6000 (Corbett) *		
C2	qPCR MasterMix No ROX (Eurogentec) Rotor-Gene ^TM^ 6000 (Corbett)		
D	Universal Probes Supermix (BioRad) Lightcycler 480 (Roche)	EvaGreen^®^ Supermix (BioRad) Lightcycler 480 (Roche)	
E	Universal Probes Supermix (BioRad) CFX96™ (BioRad) *	EvaGreen^®^ Supermix (BioRad) CFX96™ (BioRad) *	
F	Universal Probes Supermix (BioRad) CFX96™ (BioRad) *	EvaGreen^®^ Supermix (BioRad) CFX96™ (BioRad)	
G	Universal Probes Supermix (BioRad) CFX96™ (BioRad) *	EvaGreen^®^ Supermix (BioRad) CFX96™ (BioRad)	
H1			Power SYBR Green PCR Master Mix (Applied Biosystem) Applied Biosystems^®^ 7500 (Applied Biosystem) *
H2			SYBR^®^ Green Supermix (BioRad) Applied Biosystems^®^ 7500 (Applied Biosystem) *

**Table 4 jof-08-00778-t004:** Calculations of the performance criteria.

Performance Criteria	Acronyms and Calculation	Legenda	Best Performance Level (%)
Accuracy	AC = (N_TP_ + N_TN_)/N	N_TP_ N_TN_ = number of true positives and true negatives N = total number of tested sample	100
Diagnostic sensitivity	DSE = N_TP_/N^+^	N^+^ = number of samples for which the assigned value is positive (i.e., *Ceratocystis platani*-positive)	100
Diagnostic specificity	DSP = N_TN_/N^−^	N^−^ = number of samples for which the assigned value is negative (i.e., *Ceratocystis platani*-negative)	100
Repeatability	DA = (N_TP_/N)^2^ + (N_TN_/N)^2^	See above	1
Reproducibility	CO—Calculate the interlaboratory pairs sharing the same (and conforming) results and infer the percentage compared to the total number of the interlaboratory pairs		100

**Table 5 jof-08-00778-t005:** Performance parameters of Taqman, EvaGreen, and SYBR Green assays evaluated through the inference of standard curves by using different master mix/Real-Time PCR system combinations (which is identified by the different alpha-numerical codes; see legend of Table 3). The curves were generated using 7 5-fold serial dilutions (from 1:1 to 1:15.625) of plane tree gDNA obtained from wood naturally infected with *Ceratocystis platani*.

Laboratory Code ^1^	Assay	E ^2^ (%)	R˄2 ^3^	Slope ^4^	Int.
Z	Taqman	96.3	0.999	−3.413	34.041
A1	Taqman	94.9	0.999	−3.451	34.426
A3	Taqman	95.3	0.999	−3.439	33.987
A4	Taqman	101.4	0.999	−3.288	31.492
B1	Taqman	99.9	0.999	−3.323	34.102
B2	Taqman	102.3	0.999	−3.269	34.129
C1	Taqman	97.9	0.999	−3.373	29.367
C2	Taqman	101.4	0.999	−3.288	29.236
D	Taqman	79.7	0.998	−3.930	35.751
E	Taqman	96.3	0.999	−3.413	33.749
F	Taqman	94.1	0.999	−3.472	34.460
OL	EvaGreen	98.3	1.000	−3.362	33.720
D	EvaGreen	74.3	0.998	−4.143	35.968
E	EvaGreen	99.5	0.999	−3.335	33.484
F	EvaGreen	98.4	0.999	−3.360	33.967
G	EvaGreen	100.7	0.999	−3.304	33.512
OL	SYBR Green	98.8	0.999	−3.352	30.738
B	SYBR Green	99.1	0.992	−3.343	30.451
PL H1	SYBR Green	96.7	0.999	−3.404	32.052
PL H2	SYBR Green	95.8	0.999	−3.427	32.092

**^1.^** Z = OL, organizing laboratory; the following letters identify all the participating laboratories, PL; ^2^ E = PCR efficiency, 100% is the maximum theoretical value, which means perfect doubling of molecules at each cycle; ^3^ R˄2 is a measure of data linearity among technical replicates of the same and the different serial dilutions, 1 is the best fit; ^4^ the slope is the angular coefficient (m) of the equation for the standard curve (y = mx + b), 3.32 is the best fit.

**Table 6 jof-08-00778-t006:** Performance parameters—diagnostic sensitivity (DSE), diagnostic specificity (DSP) accuracy (AC) and reproducibility (CO)—of Taqman (TM), EvaGreen (EG), and SYBR Green (SG) assays inferred for each laboratory/master mix/Real-Time PCR system combination and globally for each assay. For the legend of the combinations laboratory/master mix/Real-Time-PCR system (represented by a single letter or an alphanumeric code following the chemistry acronym), see Table 3. Values of performance parameters are those calculated based on the results at the level of technical replicates, i.e., the more stringent criterion. Confidence intervals for the laboratory combinations and the assays were 96.3–100, wherever performance values showed the best fit, i.e., = 100. Where values were not best fitting confidence intervals are reported in brackets (grey-shaded).

Performance Parameters	TM.Z	TM.A1	TM.A2	TM.B1	TM.B2	TM.C1	TM.C2	TM.D	TM.E	TM.F	TM.G	TM Global
DSE	100	100	100	100	100	100	100	100	100	100	100	100
DSP	100	100	100	100	88.9 * (81.2–93.7)	100	100	100	100	100	100	99.0 (94.5–99.9)
AC	100	100	100	100	95.8 * (89.9–98.3)	100	100	100	100	100	100	99.6 (95.6–99.9)
CO												99.2 (94.8–99.9)
**Performance parameters**	**EG.Z**	**EG.D**	**EG.E**	**EG.F**	**EG.G**	**EG** **global**	**SG.Z**	**SG.B**	**SG.H1**	**SG.H2**	**SG** **global**	
DSE	100	100	100	100	100	100	100	100	100	100	100	
DSP	100	100	100	100	100	100	100	100	100	66.7 * (57.0–75.2)	91.7 (84.6–95.7)	
AC	100	100	100	100	100	100	100	100	100	87.5 * (79.6–92.6)	96.9 (91.4–98.9)	
CO						100					93.2 (86.5–96.7)	

* Statistically different value (*p*-value < 0.001).

**Table 7 jof-08-00778-t007:** Performance parameters of analytical sensitivity—diagnostic sensitivity (DSE), repeatability (DA) accuracy (AC), and reproducibility (CO)—of Taqman (TM), EvaGreen (EG), and SYBR Green (SG) inferred for each laboratory/master mix/Real-Time-PCR system and globally for each assay. For the combinations of laboratory/master mix/real-time PCR system (represented by a single letter or an alphanumeric code following the assay acronym), see Table 3. Confidence intervals for the laboratory combinations and the assays were 96.3–100 or 0.96–1 wherever performance values show the best fit, i.e., = 100 or 1. When performance values do not show a best fit, the confidence intervals are reported in bracket (grey-shaded).

Performance Parameters at 15fg ^1^	TM.Z	TM.A1	TM.A2	TM.B	TM.C	TM.E	TM.F	TM.G	TM. Global
DSE	100	100	100	100	100	100	83.3 * (74.8–89.3)	83.3 * (74.8–89.3)	95.7 (89.8–98.3)
DA	1	1	1	1	1	1	0.7 * (0.10–0.98)	0.7 * (0.10–0.98)	N.E.^2^
AC	100	100	100	100	100	100	83.3 * (74.8–89.3)	83.3 * (74.8–89.3)	95.7 (89.8–98.3)
CO									92.9 (86.1–96.5)
**Performance** **parameters at 15fg ^1^**	**EG.Z**	**EG.E**	**EG.** **global**	**SG.Z**	**SG.H1**	**SG.H2**	**SG.** **global**		
DSE	100	100	100	100	100	100	100		
DA	1	1	N.E. ^2^	1	1	1	N.E. ^2^		
AC	100	100	100	100	100	100	100		
CO			100				100		
**Performance** **parameters at** **3 fg ^1^**	**TM.Z**	**TM.A1**	**TM.A2**	**TM.B**	**TM.C**	**TM.E**	**TM.F**	**TM.G**	**TM.** **global**
DSE	100	50 * (40.3–59.6)	83.3 * (74.8–89.3)	100	100	83.3 * (74.8–89.3)	83.3 * (74.8–89.3)	100	87.2 (79.2–92.4)
DA	1	0.2 * (0.01–0.86)	0.7 * (0.1–0.98)	1	1	0.7 * (0.1–0.98)	0.7 * (0.1–0.98)	1	N.E. ^2^
AC	100	50 * (40.3–59.6)	83.3 * (74.8–89.3)	100	100	83.3 * (74.8–89.3)	83.3 * (74.8–89.3)	100	87.2 (79.2–92.4)
CO									76.2 (67.0–83.4)
**Performance** **parameters at** **3 fg ^1^**	**EG.Z**	**EG.E**	**EG.** **global**	**SG.Z**	**SG.H1**	**SG.H2**	**SG.** **global**		
DSE	100	100	100	100	100	100	100		
DA	1	1	N.E. ^2^	1	1	1	N.E. ^2^		
AC	100	100	100	100	100	100	100		
CO			100				100		

^1^ DSP is not calculated as samples with negative assigned value are not included in the analytical sensitivity test; ^2^ N.E. Not expected; * Statistically different value (*p*-value < 0.001).

## Data Availability

Not applicable.

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
