# Peer review of "Interlaboratory Performance of a Real-Time PCR Method for Detection of Ceratocystis platani, the Agent of Canker Stain of Platanus spp."

_jof, 2022, doi:10.3390/jof8080778_

Round 1
Reviewer 1 Report
The present manuscript entitled Interlaboratory performance of a Real-Time PCR method for detection of Ceratocystis platani, the agent of canker stain of Platanus spp. described a new suitable method based on Real-Time PCR that used EvaGreen, SYBR Green, and Taqman assays, to detect Ceratocystis platani (CP) in a safe mode, with interlaboratory validation. This method was previously developed for detection, validated in-house and included in the official EPPO standard PM7/14 (2).
Firstly, the authors reported on the sample preparation that was taken from fifteen plane trees. Then their phytosanitary state was established with both morphological/biological assays and Real-Time PCR and the methodology for RT-PCR was prepared and used by all nine laboratories. The inter-laboratory reproducibility by generating standard curves was determined. All nine laboratories used three assays: Eva Green, SYBR 561 Green, and Taqman, and the data obtained were used for evaluating the performance of the test: the detection of blind coded samples, the validation of the analytical sensitivity, and the assessment of the validation parameters.
Generally, the data obtained and reported in the manuscript are well corroborated, discussed, and convincingly show that Real-Time PCR technology is a gold standard for early, reliable, accurate, robust, and sensitive pathogen detection.
The manuscript is concise and the appropriate references are cited.
The authors need to address the below comments to strengthen the quality of the manuscript:
1. Title: write in italic the name of the fungus: Ceratocystis platani.
2. The manuscript could be checked for minor typo mistakes.
3. In Conclusion, do not add new information and references. These could be moved to the Discussion part. Please write the conclusions of the present study.
Author Response
Dear Referee thank you for appreciating the manuscript
here below is a point by point replay to your comments
- Title: write in italic the name of the fungus: Ceratocystis platani.
- Pilotti: Yes, we did it
- The manuscript could be checked for minor typo mistakes.
- Pilotti: Manuscript has been revised by an English proofreader
- In Conclusion, do not add new information and references. These could be moved to the Discussion part. Please write the conclusions of the present study
- Pilotti: We made this rearrangement as you required. See lines 726-735 grey-shaded.
best regards
Massimo pilotti
Reviewer 2 Report
The manuscript (Interlaboratory performance of a Real-Time PCR method for detection of Ceratocystis platani, the agent of canker stain of Platanus spp.) is interesting. It introduces clear evaluation of Real-Time PCR as diagnostic tool for C. platani detection. The method could be useful to prevent the spread of canker stain in different countries.
It is important that the English should be carefully revised.
Introduction; I think it is important to write a short paragraph about the pathogen spread and transmission methods.
Methods; L115 please specify the sampling locations.
Please correct the following points:
L49 Chane to accurate monitoring and investigating resistance level
L60 Change to Several countries in south
L80 Change to diagnostic methods is instrumental in accomplishing
L146 ground not grinded
L177 grammatical error (chooses)
L198 grammatical error detailed information
L229 grammatical error extract sample was supplied
L297 Change to the real sample status
L453 Change to the use of Thermo Scientific
L456 All basic information
L669 adjusted for the use of master mixes
L672 Grammatical error (the performance did not substantially change)
L779 Epidemiological information potentially obtainable with detection of
Author Response
Dear Referee, thank you for appreciating the manuscript.
English has been revised by an English proofreader
All corrections that you suggested have been included in the text and are yellow-shaded (numbering of lines has been changed after English revision, see below)
Here below find a point by point replay to your corrections and comments
Introduction; I think it is important to write a short paragraph about the pathogen spread and transmission methods.
M.Pilotti: Yes we did it. see lines 59-60, 62-70
Methods; L115 please specify the sampling locations.
M.Pilotti: see lines 131-132
Please correct the following points:
L49 Chane to accurate monitoring and investigating resistance level
M.Pilotti: to investigate follows to safeguard. Otherwise, the meaning of the sentence change (Lines 49-50)
L60 Change to Several countries in south
Pilotti: See line 71
L80 Change to diagnostic methods is instrumental in accomplishing
Pilotti: text has been changed after English revision (Line 88)
L146 ground not grinded:
Pilotti: See line 153
L177 grammatical error (chooses)
Pilotti: See line 186
L198 grammatical error detailed information
Pilotti: See line 208
L229 grammatical error extract sample was supplied
Pilotti: text has been changed after English revision
L297 Change to the real sample status
M.Pilotti: see line 308 (after English revision)
L453 Change to the use of Thermo Scientific
Pilotti: text has been changed after English revision
L456 All basic information
Pilotti: text has been changed after English revision
L669 adjusted for the use of master mixes
Pilotti: text has been changed after English revision. Lines 632-633
L672 Grammatical error (the performance did not substantially change)
Pilotti. See line 636
L779 Epidemiological information potentially obtainable with detection of
Pilotti: text has been changed after English revision. Lines 734-735
Best regards
M.Pilotti